# Engineering orthogonal dual transcription factors for multi-input synthetic promoters

Andreas K. Brödel[1], Alfonso Jaramillo[2,3] & Mark Isalan[1]

Synthetic biology has seen an explosive growth in the capability of engineering artificial gene circuits from transcription factors (TFs), particularly in bacteria. However, most artificial networks still employ the same core set of TFs (for example LacI, TetR and cI). The TFs mostly function via repression and it is difficult to integrate multiple inputs in promoter logic. Here we present to our knowledge the first set of dual activator-repressor switches for orthogonal logic gates, based on bacteriophage λ cI variants and multi-input promoter architectures. Our toolkit contains 12 TFs, flexibly operating as activators, repressors, dual activator–repressors or dual repressor–repressors, on up to 270 synthetic promoters. To engineer non cross-reacting cI variants, we design a new M13 phagemid-based system for the directed evolution of biomolecules. Because cI is used in so many synthetic biology projects, the new set of variants will easily slot into the existing projects of other groups, greatly expanding current engineering capacities.

[1] Department of Life Sciences, Imperial College London, London SW7 2AZ, UK. [2] School of Life Sciences, University of Warwick, Coventry CV4 7AL, UK. [3] Institute of Systems and Synthetic Biology, Genopole, CNRS, Université d'Évry, 91030 Évry, France. Correspondence and requests for materials should be addressed to M.I. (email: m.isalan@imperial.ac.uk).

Computation in living cells has the potential to revolutionize the fields of biotechnology and medicine. The main goal of biological computation is to develop devices that enable the transfer of biological information (input) into a programmable output[1]. Although such devices can be based on genetic circuits constructed at either the transcriptional or translational level the former has been more widely used to rewire living entities, and employs transcription factors (TFs) to activate or repress genes of interest. TFs are DNA-binding proteins that often function by recruiting or blocking RNA polymerase activity at gene promoters, and these functions can be combined in modular ways to engineer synthetic gene networks[2].

Many of the first bacterial synthetic gene circuits were based on a core set of only three repressors, namely TetR, LacI and bacteriophage λ cI (refs 3–6). Since then, efforts have been made to expand the set of DNA-binding proteins for the design of more complex circuits, mainly focusing on repressor molecules[7–11], although more recently several activators were developed[12,13]. Nonetheless, there are fewer activators and indeed sets of dual activator–repressor or repressor–repressor switches have not been reported thus far, even though these would be very useful for engineering devices. λ cI is an ideal candidate to address this limitation because it is one of the best-studied TFs, it functions as both a repressor and activator on a natural bidirectional promoter (λ $P_R/P_{RM}$ (ref. 14)), and structural information is available to guide re-engineering[15–17]. This knowledge can be in principle applied to construct combinatorial cI libraries with modified operator binding specificity, providing the basis for the selection of new cI variants against engineered promoters. Importantly, all the new TFs must be orthogonal to each other to maximize utility[7]. Therefore the cross-reactivity of all selected cI variants and promoter pairs needs to be ruled out before they can be applied in gene networks (Fig. 1).

To achieve the goal of an orthogonal set of TF activators (and potential repressors), we developed a new selection platform to satisfy three main requirements: (1) The process of selection has to occur inside the host cell with a high efficiency to allow rapid enrichment and to ensure compatibility with the host. (2) The system has to be compatible with the use of combinatorial libraries. (3) Selection using basally-active promoters needs to be feasible because gene networks function with background gene expression, even in the absence of an input signal. To fulfil these requirements, we developed a system based on conditional M13 bacteriophage replication. The selection process takes place inside cells by linking a selectable TF activity to conditional phage production by expressing a missing essential phage gene (Fig. 2a). In contrast to the previously reported phage-assisted continuous evolution system, known as PACE (refs 18–20), our platform was designed for the use of combinatorial library selection by packaging a phagemid (PM), while providing most phage

components on a helper phage plasmid (HP), and the selection logic on a third 'accessory' plasmid (AP) (Fig. 2a). Although this has several advantages, it also raises practical engineering requirements that are distinct from PACE. First, the smaller size of the phagemid enables the construction of combinatorial libraries with a much higher number of gene variant members than standard phage. This is useful for batch selection, although the system can also be used for continuous evolution, as long as there is a source of continuous mutation. Second, by splitting the selectable functions (PM) from the conditional gene circuit (AP), there is less chance of the evolution of 'cheaters' based on mutations upregulating the conditional missing gene. New cells, containing fresh accessory and helper plasmids, always ensure that only the selectable TF gene on the phagemid is changing. However, this raises new challenges because the choice of the missing gene becomes an important qualitative parameter: to ensure the selection of cI variants against engineered promoters, we used gene VI as the conditional gene, instead of the gene III used in PACE, because even low levels of pIII render cells resistant to phage infection[21,22]. Thus, in our system, cI-derived TFs expressed pVI to complete the phage life-cycle, allowing selective enrichment of active cI gene variants.

Overall, this work presents a set of engineered orthogonal TFs and promoters which can be used in a versatile manner, to achieve activation and dual activation-repression functions that were not previously possible. λ cI is already a member of the core set of repressors used for the first synthetic gene circuits and has been applied widely[23–29]. As a result, the set of cI variants will fit into existing engineering projects and expand the repertoire of DNA-binding proteins for gene network engineering.

## Results

**M13 phagemid-based TF selection system.** To obtain a system that would select new active TF and promoter combinations we designed a phagemid-based approach that is similar to PACE (refs 18–20), but which can handle large combinatorial libraries (Fig. 2a). The system constitutively provides all that is needed for phage production on a non-packaged helper phage plasmid (HP), except for two missing genes (gVI and gIII). A second accessory plasmid (AP) contains a conditional gene circuit that links an inducible promoter to one of the missing genes (gVI). A third plasmid, termed a phagemid (PM), because it can be packaged into an infectious phage, contains gIII and provides a combinatorially-randomized range of TFs. These TFs can potentially activate the missing gVI to complete the phage life-cycle, thus enriching those variants. In this way, by altering the promoter design, a range of new TF and promoter activities may be selected.

Early in the design phase, we hypothesized that the conditional pIII used in the PACE system leads to infection resistance when

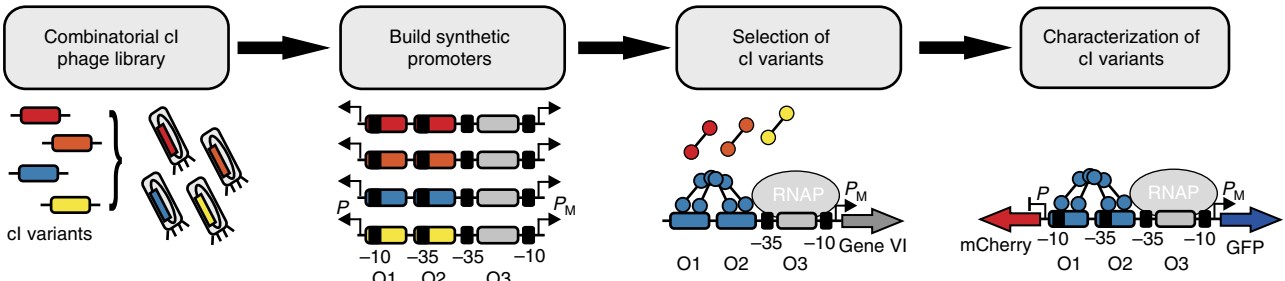

**Figure 1 | Developing a set of orthogonal dual transcription factors for synthetic logic gates.** Flow chart of the selection process and characterization of orthogonal transcription factor-promoter pairs. For the selection of cI variants, a new M13 phagemid-based system was developed (Fig. 2). The selected TFs were characterized, and checked for orthogonality, by a GFP and mCherry dual reporter assay employing bidirectional promoters that integrate both activation and repression activities.

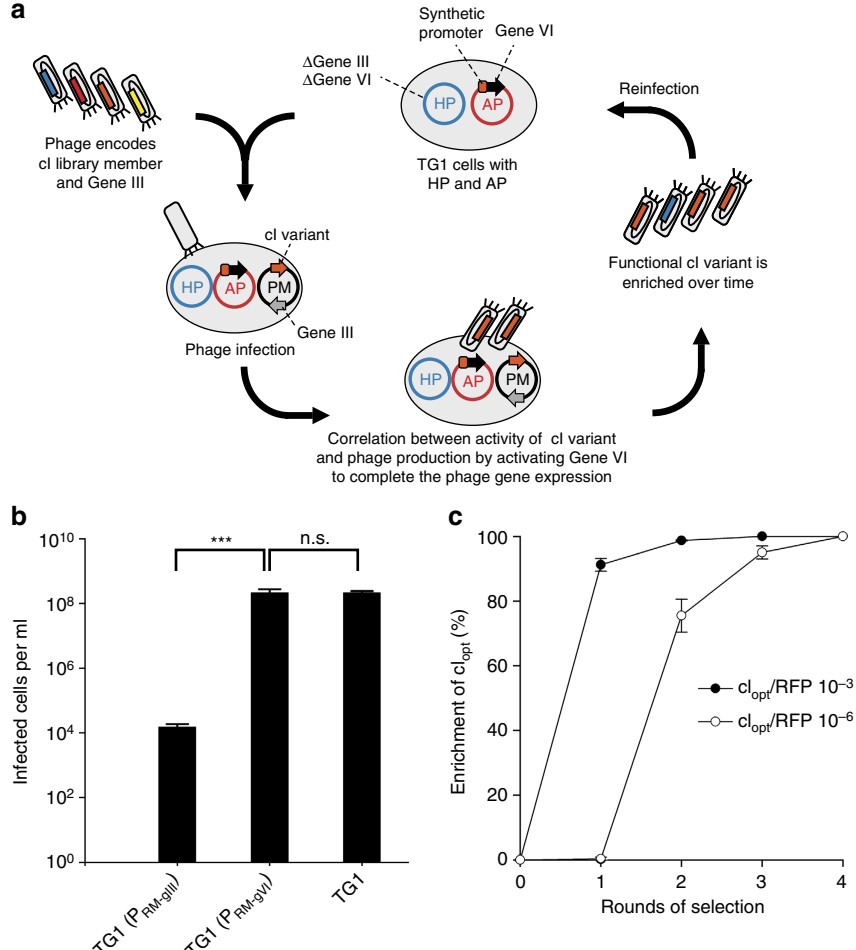

**Figure 2 | Developing a phagemid-based system for the selection of combinatorial libraries.** (**a**) Scheme of the phage-assisted selection system. *E. coli* cells containing the modified M13 helper phage (HP; contains all phage genes except genes III and VI) and an accessory plasmid (AP; containing a conditional gene VI expression circuit, dependent on variant cI activity) are infected with selection phages encoding a combinatorial cI library member on phagemids (PM; contain the variable cI genes and gene III). After infection, a protein with desired characteristics leads to an upregulated gene VI expression and therefore increased phage production. In this way, a protein with desired properties can be selected after several rounds of reinfection. (**b**) Testing for phage infection resistance from gene III or gene VI expression under the constitutive λ $P_{RM}$ promoter. TG1 cells expressing no phage genes are used as a benchmark for maximum infection potential. A *t*-test was performed to test significance (*** = $P$ value < 0.001; n.s. = not significant: $P$ value > 0.05). (**c**) Enrichment assays of λ $cI_{opt}$ from mixed phage populations, diluted $1:10^3$ or $1:10^6$ with excess of RFP-expressing phagemid. Enrichment of $cI_{opt}$ was analysed by calculating the ratio of white ($cI_{opt}$) to red (RFP) colonies on agar plates. In all experiments, phage encoding $cI_{opt}$ were fully enriched after the selection process. Error bars are 1 s.d.

expressed under a basally-active promoter[21,22]; this is problematic for the selection of bacteriophage λ cI promoter variants, which have natural basal activity and are always present on the AP in the host cells. To examine this we analysed the infection rate of TG1 cells expressing either pIII or pVI under the basally-active $P_{RM}$, the promoter of the rightward regulatory region of phage λ (Fig. 2b). The infection rate was reduced ~10,000-fold by gene III expression whereas it was not affected by gene VI. As a result, the selectable TF activity was linked to pVI production in our selection system. Gene III was instead moved onto the phagemid as this results in phage production only after initial infection and so circumvents infection resistance. This design also decreases the probability of mispackaging non-functional library members as a bystander effect from multiple infections in the same cell; after a single infection, the expression of gIII will reduce further infections[21,22]. Secondary phage are therefore less likely to act as 'cheaters' in the presence of one active TF-expressing phagemid. In addition, as pVI is unstable in the absence of

pIII (ref. 22), pVI should not accumulate before phage infection, thus reducing selection background.

The performance of the selection process was evaluated by enrichment assays. Based on the literature, we first engineered a λ cI optimized mutant ($cI_{opt}$) with a strong activation region[16]. We then tested for enrichment of $cI_{opt}$ out of both $10^3$–fold and $10^6$–fold excesses of phagemid expressing a non-TF control protein (red fluorescent protein, RFP). The $cI_{opt}$ was fully enriched within two rounds of selection from $10^3$–fold excess RFP, and in four rounds from $10^6$–fold excess RFP (Fig. 2c).

A more subtle question was whether $cI_{opt}$ would be selected against an excess of slightly-less-active wild-type (WT) cI protein; $cI_{opt}$ is described as having ~four-fold greater activity than cI (ref. 16) and has about two-fold increased activity in our hands (Supplementary Fig. 1a–c). Indeed, we enriched the stronger activator $cI_{opt}$ out of an excess of $10^3$ cI-expressing phagemid after six rounds of selection, which demonstrates the ability of the system to discriminate between differentially active TFs

(Supplementary Fig. 1d–f). These data imply that gene VI expression is linked to the target TF's activity on the phagemid and that a higher level of pVI leads to increased phage production and stronger selection.

**Construction of synthetic promoters and cI libraries**. Next, we set out to develop synthetic bidirectional promoters that were orthogonal to WT cI activity. Each bidirectional promoter $P/P_M$ consists of three operator sites named O1, O2, O3 (Fig. 3a). TF binding to O2 leads to an improved DNA-polymerase interaction and thus transcription activation of $P_M$. TF binding to O1 assists in this process as this results in an increased TF affinity to O2. TF binding to O1 also leads to repression of $P$, whereas repression of $P_M$ is obtained by O3 binding.

Synthetic promoters were designed by making symmetric variants of the consensus $\lambda$ sequence (CS) that is based on the six natural $\lambda$ operators ($O_L1$, $O_L2$, $O_L3$, $O_R1$, $O_R2$, $O_R3$) from the leftward $P_L$ and rightward $P_R$ promoters[30]. Our mutant operators differ from the CS by 2–4 base pair substitutions in the conserved penta-site, with the equivalent number of mutations in the corresponding symmetric half-site (Fig. 3a, Supplementary Fig. 2). The engineered promoters were named after the position of the base substitution in the consensus half-site (for example Fig. 3a). Each synthetic promoter consisted of the new mutant operator at position O1 and O2, whereas an inactivated O3 site was used to bypass autorepression at high cI concentrations[29]. The relative activities of the engineered $\lambda$ $P_M$ and $P$ promoters were characterized by GFP and mCherry expression in relation to $\lambda$ $P_{RM}$ and $P_R$ (Supplementary Fig. 3). All engineered $P_M$ promoters had a similar basal GFP expression, whereas mCherry expression varied between promoters because of base pair substitutions within (and next to) the $-35$ and $-10$ regions. It should be noted that the natural promoter $P_R$ is about 30-fold stronger than $P_{RM}$ (ref. 31). To ensure orthogonality, the lack of binding of WT $\lambda$ cI and $cI_{opt}$ on the designed promoters was confirmed by the reporter assay (Fig. 3b,c). Expression of cI and $cI_{opt}$ resulted in GFP activation and simultaneous mCherry

repression only on the $P_R/P_{RM}$ promoter, whereas this effect was not observed for any of the synthetic promoter variants.

The next step was to build combinatorial cI mutant libraries to target the synthetic promoter variants. Combinatorial libraries of $cI_{opt}$ were constructed by randomizing amino acids known to contact DNA in the crystal structure[32]. For example, residues in $\alpha$-helix 3, as well as at position 55 are involved in sequence-specific DNA recognition and binding to the major groove of DNA (ref. 32). Thus, we chose as the main target sites for randomization the residues Ser-45, Gly-46 and Asn-55, which make direct contacts with the operator at positions four to six. These positions were randomized in several libraries in addition to three different amino acid substitutions per library, to add extra diversity (Supplementary Tables 1 and 2).

**Selection and characterization of cI library variants**. We constructed the appropriate accessory plasmids for a combination of mutant cI selection and WT counterselection of gVI induction. Since the natural $\lambda$ $P_R/P_{RM}$ promoter contains three operators that mediate both activation and repression upon cI binding, the system naturally lends itself to both selection via activation functions and counterselection via repression. For example, the obliterated O3 site of the engineered promoters (Supplementary Fig. 2) can be replaced with the consensus WT sequence ($O_{CS}$). Thus, binding of a cI library member to mutant O1–O2 activates gene VI expression and selection, while simultaneous binding to WT O3 represses gene VI, enabling counterselection against cross-reacting WT activity (Supplementary Fig. 4c,d). This counterselection was employed on all AP to increase selection efficiency.

Libraries were selected against engineered promoters for six to eight rounds (see Methods) leading to enrichment of TFs with binding activations against their promoters. We observed that the selected TFs possessed at least four amino acid substitutions in the target region. In particular, the positions 46, 48 and 55 were substituted in all cI variants compared with wild-type cI (Fig. 4a). Interestingly, amino acid substitutions were not only obtained at

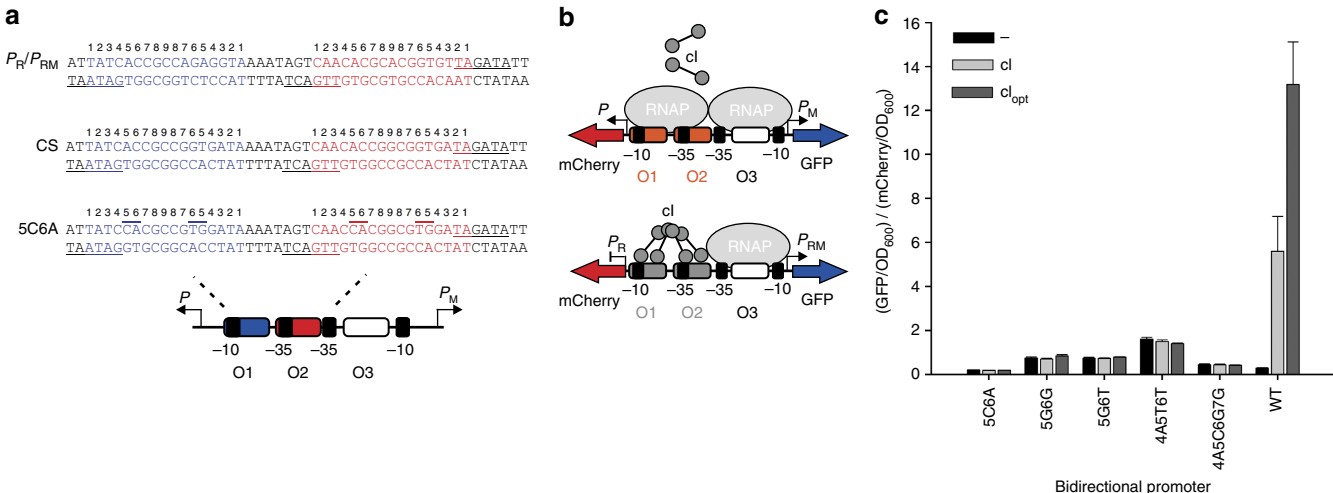

**Figure 3 | Design and characterization of engineered promoters with no activation by wild-type (WT) cI.** (**a**) Synthetic promoters were designed by making symmetric variants of the consensus sequence (CS) based on the six natural $\lambda$ operators. Each operator contains a consensus (1–8) and a non-consensus half-site (8–1), which were modified for each synthetic promoter. The engineered promoters were named after the position of the base substitution in the consensus half-site. As an example, the promoter 5C6A contains two base pair mutations at positions 5 and 6 (overlined). The $-35$ and $-10$ regions are underlined for each bidirectional promoter. (**b**) Scheme of the $\lambda$ cI binding assay with GFP and mCherry as reporter proteins. Binding of WT cI to the WT promoter ($P_R/P_{RM}$) results in activation of GFP and simultaneous repression of mCherry. (**c**) Experimental results of the binding assay. GFP and mCherry were normalized to $OD_{600}$ and the ratio was calculated. The lack of binding of $\lambda$ cI and $cI_{opt}$ against synthetic promoters was confirmed by the reporter assay. Error bars are 1 s.d.

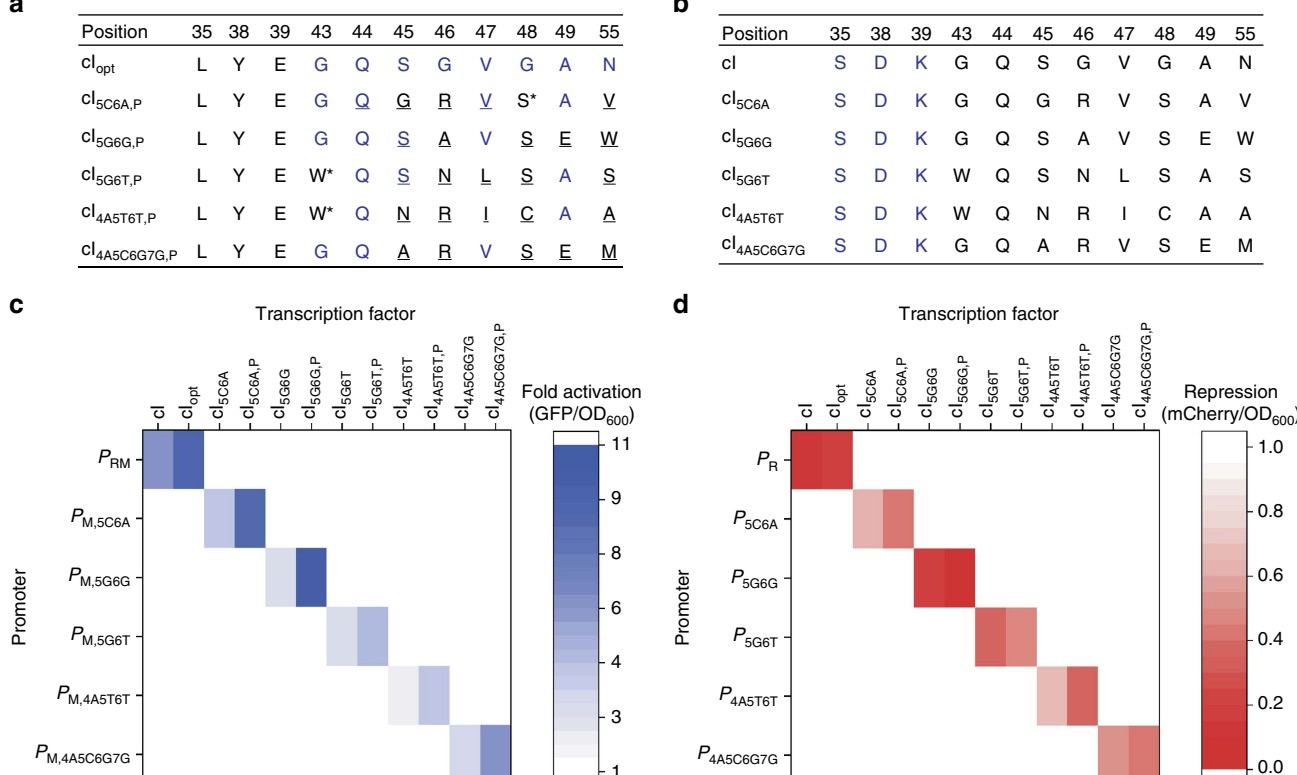

**Figure 4 | Selection and characterization of cI library members.** (**a**) Sequencing results of selected cI variants against synthetic promoters while counterselecting against wild-type binding. Wild-type amino acids are highlighted in blue, randomized positions are underlined and evolved amino acids that were not part of the combinatorial library are annotated with an asterisk. (**b**) The selected cI variants were re-engineered by rational design (35 S, 38 D, 39 K, highlighted in blue) in order to obtain activators with varying strengths. (**c**) Fold-activation of engineered λ $P_M$ promoters by selected cI variants. Stronger activation domain mutants are denoted by a 'P' (for example cI$_{5C6A,P}$). (**d**) Repression of engineered λ $P$ promoters (0.0 = 100% repression). GFP and mCherry expression was normalized to $OD_{600}$ and data were obtained from four replicates. Activation and repression were normalized to the basal expression of each promoter in the absence of a TF.

randomized positions but also spontaneously at certain positions not covered by the combinatorial space of the library. For instance, the substitution Gly to Trp at position 43, which contacts the DNA sugar-phosphate backbone of the operator between base pair 5 and 6 (ref. 32), occurred twice—independently—in the selections against $P_{M,5G6T}$ and $P_{M,4A5T6T}$. The impact of these spontaneous mutations (G43W, G48S) on protein activity was investigated by restoring the wild-type glycine amino acid at these positions (Supplementary Fig. 5). Dual activation-repression was significantly decreased for all three glycine variants, compared to the selected TFs, demonstrating the impact of the spontaneous mutations. Fixing mutations under a selection pressure is a common feature of directed evolution and can arise either from mutations during cloning or from the spontaneous error rate of M13 phage replication (~0.0046 mutation rate per genome per replication[33]).

The selected TFs were analysed for activation and repression of bidirectional promoters with a reporter assay (Supplementary Fig. 3). For baseline comparison, expression of WT cI and cI$_{opt}$ resulted in a 6-fold and 9-fold activation of $P_{RM}$, respectively, and simultaneous ~90% repression of mCherry (Fig. 4c,d). For the selected TFs, GFP expression was upregulated 4-fold to 10-fold and mCherry production was repressed simultaneously in the range 94–52%. These results demonstrate that all selected TFs are capable of dual activation and repression of their engineered bidirectional promoters.

To obtain TFs with two different activator strengths, we re-engineered the selected cI variants by rational design. We

weakened the polymerase interaction by the WT amino acid substitutions Ser-35, Asp-38 and Lys-39 (Fig. 4b). This resulted in a two-to-three-fold lower activation for each re-engineered TF (Fig. 4c). We also analysed the activation and repression of each TF against the full set of promoters in order to measure all of the possible cross-reactions, and thus establish the level of orthogonality. The data were used to construct two expression matrices that show the activity and specificity of each TF and promoter (Fig. 4c,d). The selected TFs did not cross-react with any other promoters confirming orthogonality of the developed set. Complete DNA sequences of all cI variants are listed in Supplementary Fig. 6.

**Multi-input logic gate construction and response functions.** The TFs and their synthetic operators enable the construction of a wide range of logic functions by engineering the promoter architecture. Variations of the three-operator sequence (O1–O2–O3) and position allow the construction of unidirectional or bidirectional promoters with one, two or three TF-inputs. For example, with 1-input promoters, this results in a total of 30 combinations covering a variety of logics; these include activation on $P_M$, repression on $P$ or $P_M$, and activation-repression and repression-repression on $P_M/P$ (Supplementary Fig. 4). We increased the size of the combinatorial promoter space by also designing 2-input and 3-input promoters (Supplementary Fig. 7). Promoters were experimentally tested for different operators and logics to confirm function.

Finally, we built integrated gene networks based on the orthogonal cI variants. pLITMUS-derived plasmids encoded the sensors as well as the cI variants, whereas pJPC12-based plasmids, containing the reporter GFP and mCherry, were used as outputs. We initially decreased the copy number on the pLITMUS plasmid in order to reduce the metabolic load on the cell[34] and to ensure plasmid stability. Our first multi-gene network consisted of two sensors, an integrated circuit with two cI variants operating on a bidirectional promoter and two reporter genes (Fig. 5a). The bidirectional promoter was designed for two inputs using the operator $O_{5C6A}$ at position one and two and CS at position three. The expression of cI and $cI_{5C6A}$ was linked to supplementing arabinose and IPTG, respectively. Addition of IPTG resulted in a concentration-dependent increase of GFP and decrease of mCherry, whereas supplementing arabinose accounted for a concentration-dependent decrease of GFP (Fig. 5b). Our second circuit consisted of cI and $cI_{5G6G,P}$, linked to the inducers arabinose and 3OC6-HSL, and a bidirectional

promoter for cI and $cI_{5G6G,P}$ binding. Analogous to the first gene network, expression of the TFs resulted in a concentration-dependent increase or decrease of the reporter proteins (Supplementary Fig. 8). These two circuits were further characterized using binary inputs (Supplementary Fig. 9). Our third circuit was a complex logic function, based on two unidirectional promoters with three inputs and two outputs (Fig. 5c). For this network, cells were induced with all combinations of the three inducers (0.01 mM IPTG, 0.1% arabinose, 1.0 μM 3OC6-HSL) and no inducer. As expected, this resulted in an inducer-dependent response for all inputs (Fig. 5d). Competition between activation and repression on a single promoter, by two simultaneously expressed TFs, can lead to outputs that are higher or lower than basal expression depending on the circuit design and expression parameters. These include promoter and RBS strengths of TF expression; the use of degradation tags on TFs; binding affinities of TFs; use of weak or strong cI or $cI_P$ variants. These data demonstrate that the selected

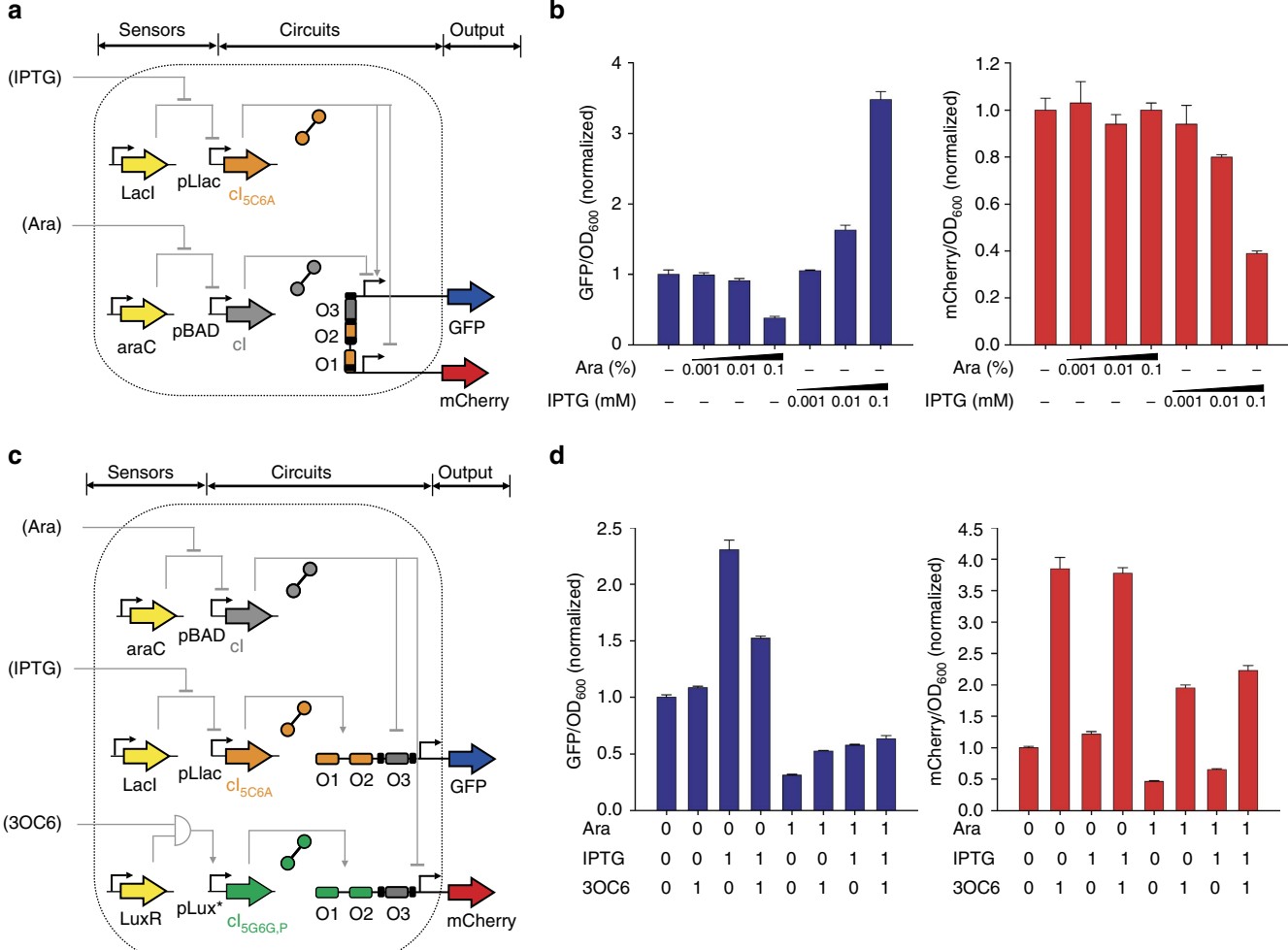

**Figure 5 | Construction and characterization of gene circuits.** (**a**) Design of a 2-input gene network. Two sensors act on an integrating circuit with two cI variants operating on a bidirectional promoter and two reporter genes. (**b**) Experimental data for the 2-input system illustrating the concentration-dependent response of GFP and mCherry. (**c**) Design of a 3-input network. This network consists of three sensors, an integrated circuit with three cI variants operating on two unidirectional promoters, and two reporter genes. (**d**) Complex logic function for the 3-input system showing the concentration-dependent correlation between inducers (Ara, IPTG, 3OC6-HSL) and output signals. The maximum GFP output is achieved by IPTG only ($cI_{5C6A}$ activates GFP expression) and the minimum GFP output by arabinose only (cI represses GFP expression). In an analogous manner, the maximum mCherry output is obtained by 3OC6-HSL ($cI_{5G6G,P}$ activates mCherry expression) and the minimum mCherry output is with Ara only (cI represses mCherry expression). As expected, combinations of the inducers resulted in intermediate GFP and mCherry expression levels, in an inducer-dependent manner. The three inducer concentrations used for the on state (1) were 0.1% Ara, 0.01 mM IPTG and 1.0 μM 3OC6-HSL. All data represent the average of four replicates and error bars correspond to 1 s.d. between the measurements.

cI variants can successfully be implemented into gene circuits and can target unidirectional as well as bidirectional promoters, in a concentration-dependent manner.

## Discussion

In this study, we expand the number of λ cI transcription activators and repressors available for use in synthetic biology and present to our knowledge the first set of dual activator–repressor and repressor–repressor switches for orthogonal logic gates. Combining simultaneous activation and repression in a single promoter enables the construction of gene circuits with new properties. It is not always easy to integrate multiple opposing inputs in synthetic promoters and the cI system presented here exploits the natural properties of λ promoters so as to obtain constructs that behave consistently. To achieve this, we developed a new phagemid-based selection system and selected new TFs against engineered promoters based on λ P and $P_M$. We then confirmed dual activation-repression and repression-repression of each TF-promoter pair and ruled out cross-reactivity to any other pair in the core orthogonal set.

We chose to engineer λ cI because of its stability, high operator specificity, and proven capability to function in gene networks as activator and/or repressor. Moreover, structural information and structure-to-function relationships are available. This enabled us to not only build cI libraries with new DNA-binding properties but also to construct cI mutants with varied activator strengths. This modification was based on the characterized relationship between amino acids involved in the RNA polymerase interaction, and could further be engineered for each member of the set in order to fine-tune the strength of activation if desired[16]. All the selected cI variants described in this work can be constructed by site-directed mutagenesis of the wild-type λ cI, enabling a quick and easy way for users to obtain the orthogonal set (Supplementary Fig. 6).

The strength of simultaneous activation and repression of the selected cI variants varied and was in the range of 4-fold to 10-fold for activation, and 94–52% for repression, which allowed us to make functional downstream logic constructs. For comparison, we obtained a six-fold activation of $P_{RM}$ in the presence of wild-type TF which is in agreement with a 5-fold to 10-fold increase reported elsewhere[16,35,36]. Variations in activation and repression can be attributed to different assay conditions as reaction parameters affect the strength of activation and repression. For example, our reporter assay was performed at 37 °C to minimize growth effects but repressive function of cI is reported to be strongest at 30 °C (ref. 37).

Promoter design was also one of the key features in this study. We engineered new unidirectional and bidirectional promoters based on λ P and $P_M$ that can be controlled by one, two or three inputs. The size of the combinatorial promoter space $N$ can be estimated by:

$$N = \sum_{k=1}^{3} \frac{n!}{k!\,(n-k)!} \times y \tag{1}$$

where $k$ is the number of inputs per promoter, $n$ is the number of orthogonal TF-operator pairs, and $y$ is the number of logics per operator combination ($y_{k=1} = 5$, $y_{k=2} = 8$, $y_{k=3} = 6$; Supplementary Fig. 10). This results in a total size of 270 promoters with one input ($N = 30$, $n = 6$), two inputs ($N = 120$, $n = 6$) or three inputs ($N = 120$, $n = 6$). As examples, a series of promoters were constructed and activation and/or repression were experimentally confirmed in each case. Two of the characterized promoters were then successfully applied in test gene networks. The expressed TFs did not show any cross-reactivity with any off-target operator and were compatible with

the inducible systems based on lacI, araC and luxR. Overall, promoter construction can be achieved in a quick and easy manner by PCR amplification of oligonucleotides. More generally, our promoter design may assist in the development of new promoters with desired properties in future studies.

The selected set of orthogonal TFs demonstrates the performance of the developed M13 phagemid-based system for the directed evolution of new biomolecules. One key advantage over conventional systems is the selection of combinatorial libraries in vivo. This ensures compatibility with the host cell machinery and applies an intrinsic selection for functional orthogonality, by disfavouring deleterious cross-reactions with the host genome. Furthermore, the cell-based evolution approach enables the exploration of a gene's sequence space wherever semi-rational design is feasible, which means that proteins can be modified based even on a partial understanding of the structural consequences of a set of changes. The selection system itself is not limited by the number of variants but rather the critical step is usually in obtaining sufficient transformant clones ($10^6$–$10^{10}$, depending on the method)[38]. By comparison, in PACE the system is not limited by transformation but by the effective mutation rate and continuous selection parameters[18]. Nonetheless, contemporary gene assembly protocols (for example, Gibson Assembly, Golden Gate Assembly[39,40]) simplify and speed up the process of library construction over traditional restriction enzyme cloning, thus reducing the number of preparation steps before selection.

The selection process itself is performed here in batch mode, which enables the performance of multiple selections in parallel and allows for straightforward scalability of each individual selection and easy handling. However, in principle, the system would be compatible with continuous flow selection and evolution methods[18]. Moreover, the platform is capable of evolving any gene on the phagemid that can be linked to pVI production, analogous to previous uses of phage-assisted evolution[18]. A wide variety of functions including protein-protein interactions[41], protease activity[20] and gene editing tools[42] have been linked to conditional M13 phage production, demonstrating the widespread applicability of such techniques. Overall, the system developed here further emphasizes the use of directed evolution as a powerful tool for the engineering of proteins with desired properties for synthetic biology applications.

## Methods

**Strains and media.** Standard DNA cloning was performed with chemically competent TOP10 cells (Invitrogen) and TG1 cells (Zymo Research). Combinatorial library cloning was performed with NEB 5-alpha electrocompetent cells or electrocompetent TG1 cells. Selection phage production was carried out with BL21(DE3) cells (NEB). All phage-assisted selection experiments and reporter assays were performed with TG1 cells. Genotypes of all strains are listed in Supplementary Table 3. Cells were grown in Luria-Bertani medium (LB: 10 g l$^{-1}$ Bacto-tryptone, 5 g l$^{-1}$ yeast extract, 10 g l$^{-1}$ NaCl), M9 minimal medium (6.8 g l$^{-1}$ Na$_2$HPO$_4$, 3.0 g l$^{-1}$ KH$_2$PO$_4$, 0.5 g l$^{-1}$ NaCl, 1.0 g l$^{-1}$ NH$_4$Cl, 2 mM MgSO$_4$, 100 µM CaCl$_2$, 0.2% (w/v) glucose, 1 mM thiamine-HCl), 2 × TY medium (5 g l$^{-1}$ NaCl, 10 g l$^{-1}$ yeast extract, 16 g l$^{-1}$ tryptone) or S.O.C. medium (Sigma-Aldrich). Chloramphenicol (25 µg ml$^{-1}$), kanamycin (50 µg ml$^{-1}$) and ampicillin (100 µg ml$^{-1}$) were added where appropriate.

**Cloning and plasmid construction.** Subcloning was carried out using either restriction sites or Gibson Assembly[39]. The weak M13 packaging signal (PS), gene III and gene VI were removed from the M13KO7 helper phage (HP-ΔPS-ΔgIII-ΔgVI). The pJPC12 vector was obtained from Peterson and Phillips[43] and the M13 packaging signal was deleted from the vector backbone. The higher copy number version pJPC13-ΔPS was obtained by site-directed mutagenesis according to Peterson and Phillips[43]. For the construction of gene circuits, a pLITMUS derivative with a lower copy number was cloned by substituting the origin of replication pUC to p15A. A modified version of the pLux promoter (pLux*) (ref. 1) was used to reduce basal expression, and a degradation tag (AANDENYALVA) was fused to cI$_{5G6G,P}$ at the C-terminal site to decrease the protein level[34] in the

absence of 3OC6-HSL. Green fluorescent protein (GFP, GenBank no. KM229386), mCherry (Uniprot no. X5DSL3) and red fluorescent protein (RFP, GenBank no. AID49074) were used as reporters. The araC-pBAD cassette was used in a previous study[34]. The stronger activator λ cI$_{opt}$ (ref. 16) was obtained via site-directed mutagenesis of three amino acids in the cI gene (GenBank no., X00166) at position 35–39 (SVADK to LVAYE). To obtain TFs with two different activator strengths, we re-engineered the selected cI variants by site-directed mutagenesis using the primers cI-F2 and cI-R2. Single amino acid mutations at positions 43 and 48 (cI$^{G48}_{5C6A,P}$, cI$^{G43}_{5G6T,P}$ and cI$^{G43}_{4A5T6T,P}$) were made by site-directed mutagenesis. Promoters, ribosomal binding sites and terminators were ordered as oligonucleotides (Sigma-Aldrich) and were obtained from previous studies[34,44]. The $P_{RM}$ promoter contained the mutated O3 sites TATAAATAGTGGTGATA (ref. 29) or ACAAACTTTCTTGTATA in order to bypass autorepression at high cI concentrations. Plasmids were purified using the QIAprep Spin Miniprep Kit or the HiSpeed Plasmid Maxi Kit (QIAGEN). Nucleotide sequences of all cloned constructs were confirmed by DNA sequencing (GATC Biotech). A set of plasmids for the selection system and the selected cI variants, along with maps and sequences, were deposited in Addgene (IDs are listed in Supplementary Table 4). All plasmids and primer sequences are listed in Supplementary Tables 4,5 and Supplementary Data 1.

**Construction of combinatorial λ cI libraries.** Combinatorial libraries were cloned based on forward and reverse primers containing NNS codons (where S = G/C) at the randomized positions (Supplementary Table 6). Primers were fused by PCR and fragments were cloned into the linearized pLITMUS-cI$_{opt}$-gIII vector by Gibson Assembly. Randomized positions of λ cI$_{opt}$ were as follows: Library 1 (44Q, 45S, 46G, 47V, 55N); Library 2 (45S, 46G, 47V, 48G, 55N); Library 3 (45S, 46G, 48G, 49A, 55N). This results in a combinatorial space of $3.2 \times 10^6$ variants per library. Cells were transformed and plated on 24 cm$^2$ Nunc BioAssay Dishes (Thermo Scientific). Transformation efficiency was estimated by colony counting of plated serial dilutions ($>10^7$ colonies per library; $>3$-fold excess). The next day, colonies were harvested and phagemid DNA was purified. All combinatorial libraries and corresponding sequencing results are listed in Supplementary Tables 1 and 2.

**Selection phage production.** Selection phage production was performed in BL21(DE3) cells containing HP-ΔPS-ΔgIII-ΔgVI and pJPC13-ΔPS-T7-gVI. Cells were made electrocompetent, phagemids transformed and cells were grown overnight at 30 °C, 250 r.p.m. (Stuart Shaking Incubator SI500) in 2 × TY medium supplemented with kanamycin, chloramphenicol and ampicillin. For enrichment assays, plasmids containing λ cI$_{opt}$ and RFP were mixed in a ratio of $10^{-3}$ and $10^{-6}$ before transformation. 0.25 mM Isopropyl β-D-1-thiogalactopyranoside (IPTG) was added to the culture after phagemid transformation to induce gene VI expression. Samples were centrifuged for 10 min at 8,000 g and supernatants were sterile filtered (0.22 μm pore size, Millex-GV). Phage concentration was analysed by TG1 infection of diluted phage stocks and colony counting on ampicillin plates.

**Gene III resistance experiment.** TG1 cells containing gene III or gene VI under the constitutive $P_{RM}$ promoter were infected at OD 0.4 and a multiplicity of infection (MOI) of 25. Samples were incubated for 1 h before serial dilutions were plated on ampicillin plates. The next day, the number of colonies were correlated to the number of phage-infected cells. TG1 cells in the absence of any plasmid were used as control.

**Phage-assisted selections.** TG1 cells containing the modified helper phage HP-ΔPS-ΔgIII-ΔgVI and the appropriate accessory plasmid were grown on M9 minimal medium plates supplemented with chloramphenicol and kanamycin. Starter cultures were grown for 6-8 h until the OD$_{600}$ reached 0.4–0.8. 5–10 ml of the starter cultures were infected with selection phages at a MOI of 0.1–10. Samples were kept at 37 °C without stirring for 10 min before they were incubated for 18–20 h at 30 °C and 250 r.p.m (Stuart Shaking Incubator SI500). Overnight cultures were centrifuged for 10 min at 8,000 g and the phage supernatant was used to start a new round of selection. After four to eight rounds of selection, phage supernatants were sterile filtered and diluted, before infecting TG1 cells containing the appropriate reporter. Infected cells were selected on ampicillin plates and at least three colonies per selection were grown overnight in LB medium. Phagemid DNA was purified using the QIAprep Spin Miniprep Kit (QIAGEN) and analysed by sequencing. For the enrichment assays, TG1 cells were infected with phage dilutions after each round of selection and plated on ampicillin plates. In the case of cI$_{opt}$/RFP selections, the ratio of white to red colonies was analysed by colony counting and white colonies were linked to cI$_{opt}$ infection by colony-PCR. In the case of cI$_{opt}$/cI selections, ten colonies were analysed by colony-PCR using the forward primer pLITMUS-F1 and the reverse primers cI$_{opt}$-R2 or cI-R3. Cross-binding of the reverse primers between the two cI variants was initially tested and ruled out.

**Reporter assay.** TG1 cells were transformed with pJPC12-based reporter plasmids and the appropriate phagemid and selected overnight on agar plates. The next day,

single colonies were picked for each biological replicate and grown 4–6 h in 1 ml 2 × TY supplemented with 5 μg ml$^{-1}$ chloramphenicol and 5 μg ml$^{-1}$ carbenicillin. The cultures were diluted to OD$_{600}$ 0.01 and 150 μl were added to the wells of a 96-well plate. The absorbance at 600 nm, green fluorescence (excitation: 485 nm, emission: 520 nm) and red fluorescence (excitation: 585 nm, emission: 625 nm) were measured every 10 min in a Tecan Infinite M200 plate reader (37 °C, shaking between readings) until the E. coli cells reached stationary phase. For data analysis, fluorescence readings in the mid-exponential phase (OD$_{600}$ of 0.2) were used. Both absorbance and fluorescence were background corrected. The fluorescence was then normalized for the number of cells by dividing by the absorbance. The average of four replicates and the corresponding s.d. was calculated for each sample.

**Construction and analysis of gene circuits.** Single colonies from TG1 cells containing the reporter plasmid and the respective p15A-based pLITMUS construct were grown for 3–4 h in 1 ml 2 × TY supplemented with 5 μg ml$^{-1}$ chloramphenicol, 5 μg ml$^{-1}$ carbenicillin and 0.5% glucose. The cultures were diluted to OD$_{600}$ 0.01 in a total volume of 150 μl in each well of a 96-well plate. IPTG (0.001, 0.01, 0.1 mM), arabinose (0.001, 0.01, 0.1%) and N-(β-ketocaproyl)-L-homoserine lactone (3OC6-HSL, 0.01, 0.1, 1.0 μM) were added where appropriate. For the 3-input circuit, cells were induced with all combinations of the three inducers (0.01 mM IPTG, 0.1% arabinose, 1.0 μM 3OC6-HSL) or no inducer. The absorbance at 600 nm, green fluorescence (excitation: 485 nm, emission: 520 nm) and red fluorescence (excitation: 585 nm, emission: 625 nm) were measured every 10 min in a Tecan Infinite M200 plate reader (37 °C, shaking between readings) until the E. coli cells reached stationary phase. For data analysis, fluorescence readings in the late-exponential phase (OD$_{600}$ of 0.5) were used. Both absorbance and fluorescence were background corrected. The fluorescence was then normalized for the number of cells by dividing by the absorbance. The average of four replicates and the corresponding s.d. was calculated for each sample.

**Data availability.** Sequences and plasmid maps (as listed in Supplementary Table 4), as well as DNA plasmids have been deposited and are available from Addgene: https://www.addgene.org/. All other relevant data are available from the corresponding author on request.

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

## Acknowledgements

This research was supported by the European Commission grant FP7-ICT-2013-10 (no 610730, EVOPROG). A.J. was funded by FP7-KBBE (no 613745, PROMYS). M.I. is funded by New Investigator award no WT102944 from the Wellcome Trust U.K. We would like to thank Marc Sturrock, Daniel Blicher Holst Hansen, Jennifer Shenton, Gabriella Santosa and Rebecca Powell for their support. We also thank Robert Bradley for providing the LuxR-pLux* cassette.

## Author contributions

Conceived and designed the experiments: A.K.B., A.J. and M.I. Performed the experiments: A.K.B. Analysed the data: A.K.B., M.I. Contributed reagents, materials or analysis tools: M.I. and A.J. Wrote the manuscript: A.K.B. and M.I.

## Additional information

**Competing financial interests:** The authors declare no competing financial interests.

