## [Peer Review File · Nature Communications]

1 Reviewer #1 (Remarks to the Author)

2

3 A. Summary of the key results

The authors provide a set of novel orthogonal transcription factors derived from variants of the cI
lambda repressor protein and its associated bidirectional promoter region, and present the
methodology used to obtain the new TFs.

B. Originality and interest: if not novel, please give references

The protein/promoter pairs offer a substantial new set of orthogonal methods for transcriptional
control, and the methodology suggests a practical method to obtain new variants in future. The
ability to provide simultaneous activation and repression in the two directions of the bidirectional
promoter is a novel contribution offering interesting design possibilities for future systems.

C. Data & methodology: validity of approach, quality of data, quality of presentation

The methodology is well described and the data is thorough and well discussed. A significant
amount of experimental work is represented here, and in general the paper presents its results
clearly and uses valid approaches.

One exception to the general clarity is the labeling of Figure 5b, where there seems to be a
mistake in the numbers associated with the percentages and concentrations of arabinose and
IPTG: many columns show as 0.0, making it difficult to understand whether concentrations are
increasing left to right, or if not, what the order of the columns does reflect.

D. Appropriate use of statistics and treatment of uncertainties

Appropriate analysis is shown.

E. Conclusions: robustness, validity, reliability

The conclusions seem well supported by the data.

F. Suggested improvements: experiments, data for possible revision

--

G. References: appropriate credit to previous work?

Yes.

H. Clarity and context: lucidity of abstract/summary, appropriateness of abstract, introduction and
conclusions

Clearly presented and summarized.

Reviewer #2 (Remarks to the Author)

The paper by Brodel et al reports the generation of a series of cI protein variants with new
specificities towards different target promoters –and their rewiring to originate a suite of genetic
circuits with a predetermined input/output logic. To this end, Authors adopted a directed evolution
strategy inspired in the method formerly developed in Liu's Lab known as Phage-Assisted
Continuous Evolution (PACE). Brodel et al refine and adapt PACE to their own needs and use
progressive phage infectivity as the readout of altered cI binding to promoters with modified DNA

sequences. In reality, I think that the main value of this work is precisely having adapted PACE to
TF evolution -an stratagem that could have a considerable fundamental and biotechnological
impact.

I have enjoyed reading this ms. The question at stake is clearly stated and the wet experimental
strategy (the main contribution of this work) is brilliant. Furthermore, the experiments were
carried out elegantly and the results (and their interpretation) quite convincing. The paper delivers
a suite of cl variants/promoters that add to the contemporary toolbox of regulatory parts for
designing novel genetic circuits.

The weak points include that [i] the phage-based evolution system that Authors adapt to their
purposes is not entirely novel (at the end of the day, it is a variant of PACE) and [ii] Authors
overdo their motivation based on the dearth of TFs available for making synthetic circuits (lines
43-46). In reality there is a large number of both repressors and activators to this end.

Reviewer #3 (Remarks to the Author)

The authors describe the use of a phagemid system for the in vivo selection of a set of
transcription factors (TFs) with orthogonal DNA recognition sequences in bacteria. Their results
indicate that each of the evolved TFs can activate and repress reporter expression only when the
correct operators are present and that this method enables the construction of complex networks.
The key innovations of this method relative to the PACE method developed in David Liu's lab lie in
the use of pVI instead of pIII to drive selection (reduced infection resistance) and in the use of the
phagemid/helper phage pair (increased library size). While PACE excels at simulating the
conditions of evolution, the method described here is much more suitable for interrogating the
relative activities of a specific set of variants. Combining this approach with deep sequencing
methods should prove valuable to many researchers. The manuscript flows logically and clearly
and there are no major concerns. Here are some minor considerations:

1) The overall clarity of the manuscript could be improved in a few places.

The section beginning on Line 139 could use a clear description of how the native and synthetic
promoters would be expected to function based on the TF occupancy of the operator sites. Does
binding to O3 always lead to repression? It is also not immediately obvious that the O3 site was
inactivated for some of the controls (or why this was done).

The purpose of pLITMUS plasmid (line 222) should be mentioned.

The size of the tested libraries is not indicated and should be.

2) Several additional experiments could strengthen the results but should not be considered
essential.

The effect of removing the Gly43Trp mutation acquired during selection would demonstrate its
importance. This could be done for all of the mutations selected in the TF panel to demonstrate
that each is critical to the selected function.

The effect of multiple inputs on the function of the two-input sensors (Fig 5b & Supp Fig 5b) would
more fully characterize their function. Moving the titrations in Fig 5b to the supplemental and
adding a binary input table similar to Fig 5d could most easily accomplish this. This could also be
performed for the engineered promoter series in Supp Fig 4.

3) A more detailed analysis of the function of the sensor circuits and a discussion of how this
method improves on existing ones will strengthen the authors' claims.

What is the practical limit for library sizes tested using this method and how does this compare
with PACE? It would be helpful to emphasize the type of applications that this system is better
suited to.

Is there an explanation for why some of the "P" TF variants in Fig 4d are more potent repressors
than their non-"P" counterparts?

Does the circuit in Fig 5c/d function as one would predict? Why does IPTG+3OC6 yield less GFP

activity relative to IPTG alone? Ara+3OC6 clearly reduces mCherry expression relative to 3OC6
alone, but why is this level of expression greater than basal levels? Is the function of networks
built from the evolved TFs predictable ab initio?

*Please note: line numbers refer to marked-up PDF manuscript.*

**Responses to the comments of Reviewer #1:**

Reviewer #1 (Remarks to the Author):

118 A. Summary of the key results

The authors provide a set of novel orthogonal transcription factors derived from variants of the cl
lambda repressor protein and its associated bidirectional promoter region, and present the
methodology used to obtain the new TFs.

B. Originality and interest: if not novel, please give references

The protein/promoter pairs offer a substantial new set of orthogonal methods for transcriptional
control, and the methodology suggests a practical method to obtain new variants in future. The ability
to provide simultaneous activation and repression in the two directions of the bidirectional promoter is
a novel contribution offering interesting design possibilities for future systems.

C. Data & methodology: validity of approach, quality of data, quality of presentation

The methodology is well described and the data is thorough and well discussed. A significant amount
of experimental work is represented here, and in general the paper presents its results clearly and
uses valid approaches.

**One exception to the general clarity is the labeling of Figure 5b, where there seems to be a
mistake in the numbers associated with the percentages and concentrations of arabinose and
IPTG: many columns show as 0.0, making it difficult to understand whether concentrations are
increasing left to right, or if not, what the order of the columns does reflect.**

*We thank Reviewer 1 for the comments. We revised Figure 5b, 5d and Suppl. Fig. 5b (Suppl. Fig. 6 in
the revised MS) to improve the general clarity of the presented data (concentration gradients are
highlighted, 0.0s are removed, binary data are now in binary order). We also included similarly-
formatted additional data for the binary 2-input systems to strengthen the results, so as to respond to
Referee 3 (please see Suppl. Fig. 7).*

D. Appropriate use of statistics and treatment of uncertainties

Appropriate analysis is shown.

E. Conclusions: robustness, validity, reliability

The conclusions seem well supported by the data.

F. Suggested improvements: experiments, data for possible revision

--

G. References: appropriate credit to previous work?

Yes.

H. Clarity and context: lucidity of abstract/summary, appropriateness of abstract, introduction and
conclusions

Clearly presented and summarized.

**Responses to the comments of Reviewer #2:**

The paper by Brodel et al reports the generation of a series of *cl* protein variants with new specificities
towards different target promoters –and their rewiring to originate a suite of genetic circuits with a
predetermined input/output logic. To this end, Authors adopted a directed evolution strategy inspired
in the method formerly developed in Liu's Lab known as Phage-Assisted Continuous Evolution
(PACE). Brodel et al refine and adapt PACE to their own needs and use progressive phage infectivity
as the readout of altered *cl* binding to promoters with modified DNA sequences. In reality, I think that
the main value of this work is precisely having adapted PACE to TF evolution -an stratagem that
could have a considerable fundamental and biotechnological impact.

I have enjoyed reading this ms. The question at stake is clearly stated and the wet experimental
strategy (the main contribution of this work) is brilliant. Furthermore, the experiments were carried out
elegantly and the results (and their interpretation) quite convincing. The paper delivers a suite of *cl*
variants/promoters that add to the contemporary toolbox of regulatory parts for designing novel
genetic circuits.

**The weak points include that [i] the phage-based evolution system that Authors adapt to their**
**purposes is not entirely novel (at the end of the day, it is a variant of PACE) and [ii] Authors**
**overdo their motivation based on the dearth of TFs available for making synthetic circuits**
**(lines 43-46). In reality there is a large number of both repressors and activators to this end.**

*We thank Reviewer 2 for the supportive comments. We revised our statement regarding available TFs*
*for making synthetic circuits and added two new references (12, 13) to the manuscript:*

*Line 47-52:* Since then, efforts have been made to expand the set of DNA-binding proteins for the
design of more complex circuits, mainly focusing on repressor molecules⁷⁻¹¹, although
more recently several activators were developed^{12,13}. Nonetheless, there are fewer
activators and indeed dual activator-repressor or repressor-repressor switches have
not been reported thus far, even though these would be very useful for engineering
devices.

**Responses to the comments of Reviewer #3:**

The authors describe the use of a phagemid system for the in vivo selection of a set of transcription
factors (TFs) with orthogonal DNA recognition sequences in bacteria. Their results indicate that each
of the evolved TFs can activate and repress reporter expression only when the correct operators are
present and that this method enables the construction of complex networks. The key innovations of
this method relative to the PACE method developed in David Liu's lab lie in the use of pVI instead of
pIII to drive selection (reduced infection resistance) and in the use of the phagemid/helper phage pair
(increased library size). While PACE excels at simulating the conditions of evolution, the method
described here is much more suitable for interrogating the relative activities of a specific set of
variants. Combining this approach with deep sequencing methods should prove valuable to many
researchers. The manuscript flows logically and clearly and there are no major concerns. Here are
some minor considerations:

*We thank Reviewer 3 for the comments. We performed additional experiments and revised the*
*manuscript according to the helpful experimental suggestions. We feel that the manuscript has been*
*improved because of these revisions.*

**1) The overall clarity of the manuscript could be improved in a few places. The section**
**beginning on Line 139 could use a clear description of how the native and synthetic promoters**
**would be expected to function based on the TF occupancy of the operator sites. Does binding**
**to O3 always lead to repression?**

*We inserted a more detailed description of how the promoters function based on TF occupancy.*

*Line 147-151: Each bidirectional promoter P/P_M consists of three operator sites named O1, O2, O3*
*(Fig. 3a). TF binding to O2 leads to an improved DNA-polymerase interaction and*
*thus transcription activation of P_M. TF binding to O1 assists in this process as this*
*results in an increased TF affinity to O2. TF binding to O1 also leads to repression of*
*P whereas repression of P_M is obtained by O3 binding.*

**It is also not immediately obvious that the O3 site was inactivated for some of the controls (or**
**why this was done).**

*Line 158-160: Each synthetic promoter consisted of the new mutant operator at position O1 and O2,*
*whereas an inactivated O3 site was used to bypass autorepression at high cl*
*concentrations²⁹.*

**The purpose of pLITMUS plasmid (line 222) should be mentioned.**

*We now mention the purpose of pLITMUS and pJPC12-derived plasmids for the construction of our*
*gene networks.*

*Line 243-245: pLITMUS-derived plasmids encoded the sensors as well as the cl variants, whereas*
*pJPC12-based plasmids, containing the reporter GFP and mCherry, were used as*
*outputs.*

**The size of the tested libraries is not indicated and should be.**

*We now mention the size of all libraries in the manuscript and in Suppl. Table 2.*

*Line 410-414: This results in a combinatorial space of 3.2×10^6 variants per library. Cells were*
*transformed and plated on 24 cm² Nunc BioAssay Dishes (Thermo Scientific).*
*Transformation efficiency was estimated by colony counting of plated serial dilutions*
*(more than 10^7 colonies per library; >3-fold excess).*

**2) Several additional experiments could strengthen the results but should not be considered**
**essential.**

**The effect of removing the Gly43Trp mutation acquired during selection would demonstrate its**
**importance. This could be done for all of the mutations selected in the TF panel to**
**demonstrate that each is critical to the selected function.**

*We performed these additional experiments to strengthen the results. We investigated the impact of*
*the single amino acid mutations which were not randomized in our combinatorial libraries but were*
*obtained spontaneously during the selection process (e.g. Gly43Trp, Supplementary Fig. 3). We*
*consequently revised the manuscript; lists of plasmids and primers were updated in the*
*Supplementary Information.*

*Line 199-203:* The impact of these spontaneous mutations (G43W, G48S) on protein activity were
investigated by restoring the wild-type glycine amino acid at these positions
**(Supplementary Fig. 3)**. Dual activation-repression was significantly decreased for
all three glycine variants, compared to the selected TFs, demonstrating the impact of
the spontaneous mutations.

*Line 392-394:* Single amino acid mutations at positions 43 and 48 ($cl_{5C6A,P}^{G48}$, $cl_{5G6T,P}^{G43}$ and
$cl_{4A5T6T,P}^{G43}$) were made by site-directed mutagenesis.

**The effect of multiple inputs on the function of the two-input sensors (Fig 5b & Supp Fig 5b)**
**would more fully characterize their function. Moving the titrations in Fig 5b to the**
**supplemental and adding a binary input table similar to Fig 5d could most easily accomplish**
**this. This could also be performed for the engineered promoter series in Supp Fig 4.**

*We performed additional experiments (new Supplementary Fig. 7) for the two-input sensors in Fig. 5*
*and old Suppl. Fig. 5 (now numbered as Suppl. Fig. 6), to further characterize their function. We kept*
*the titration in the Fig. 5b because we'd like to show a TF concentration-dependent effect on*
*activation-repression in the MS and not only binary inputs. Dual input experiments for old Suppl. Fig.4*
*(now 5) cannot be performed, as the TFs are on separate plasmids and thus cannot be expressed*
*simultaneously. This would require further cloning. The manuscript was revised as follows:*

*Line 257-258:* These two circuits were further characterized using binary inputs **(Supplementary**
**Fig. 7)**.

**3) A more detailed analysis of the function of the sensor circuits and a discussion of how this**
**method improves on existing ones will strengthen the authors' claims.**

*The following section was included in the discussion:*

*Line 276-280:* Combining simultaneous activation and repression in a single promoter enables the
construction of gene circuits with new properties. It is not always easy to integrate
multiple opposing inputs in synthetic promoters and the cl system presented here
exploits the natural properties of λ promoters so as to obtain constructs that behave
consistently.

**What is the practical limit for library sizes tested using this method and how does this**
**compare with PACE?**

*We inserted a statement regarding the practical limit of the method and how this compares with*
*PACE.*

*Line 328-331:* The selection system itself is not limited by the number of variants but rather the
critical step is usually in obtaining sufficient transformant clones (10^6 - 10^{10} , depending
on the method)³⁸. By comparison, in PACE the system is not limited by transformation
but by the effective mutation rate and continuous selection parameters¹⁸.
Nonetheless, contemporary gene assembly protocols (e.g., Gibson Assembly, Golden
Gate Assembly^{39,40}) simplify and speed up the process...

**It would be helpful to emphasize the type of applications that this system is better suited to.**

*The type of applications that this system is better suited to than PACE are mentioned in the*
*manuscript such as for selections against basally-active promoters (line 68-70), for selections where*
*semi-rational design is feasible (line 325-326) and for selections in batch mode (line 336-338).*

*Line 68-70:* Selection using basally-active promoters needs to be feasible because gene networks
function with background gene expression, even in the absence of an input signal.

*Line 325-326:* ... the exploration of a gene's sequence space wherever semi-rational design is
feasible, ...

*Line 336-338:* The selection process itself is performed here in batch mode, which enables the
performance of multiple selections in parallel and allows for straightforward scalability
of each individual selection and easy handling.

**Is there an explanation for why some of the "P" TF variants in Fig 4d are more potent**
**repressors than their non-"P" counterparts?**

*We assume that "P" variants generally possess a stronger polymerase interaction so there are likely*
*less polymerases per cell available for expression of mCherry under the P promoter. This competition*
*may lead to decreased mCherry concentrations even with the same strength of TF-DNA interaction.*
*Alternatively, the stronger polymerase interaction may lead to the polymerase providing some DNA-*
*binding energy to the interaction, effectively increasing the affinity of the repressor and thus yielding*
*higher repression. These "P" repression effects are weaker than the activation properties, and are*
*more complex, so we do not emphasise them in the manuscript.*

**Does the circuit in Fig 5c/d function as one would predict? Why does IPTG+3OC6 yield less**
**GFP activity relative to IPTG alone?**

*Yes, the circuit functions as one would predict. IPTG+3OC6 yield less GFP activity relative to IPTG*
*alone because the total number of polymerases per cell are split between these two processes*
*leading to an expected lower GFP expression for these two inputs.*

**Ara+3OC6 clearly reduces mCherry expression relative to 3OC6 alone, but why is this level of**
**expression greater than basal levels? Is the function of networks built from the evolved TFs**
**predictable ab initio?**

*The functions are predictable for activation or repression. However, for functions with competition*
*between activation and repression on a single promoter, the outputs cannot be easily predicted as*
*this depends on multiple parameters such as: 1) Promoter and RBS strengths of TF expression 2)*
*Use of degradation tags on TFs 3) Binding affinities of TFs 4) Which variant (cl or cl_P) is used in the*
*circuit. This results in an output that can be higher or lower than basal expression levels depending on*
*the design. We included two sentences in the manuscript to address this issue:*

*Line 262-267:* Competition between activation and repression on a single promoter, by two
simultaneously expressed TFs, can lead to outputs that are higher or lower than basal
expression depending on the circuit design and expression parameters. These
include promoter and RBS strengths of TF expression; the use of degradation tags on
TFs; binding affinities of TFs; use of weak or strong cl or cl_P variants.

Reviewers' Comments:

Reviewer #3 (Remarks to the Author)

I thank the authors for their thorough response; the updates to the manuscript satisfactorily
addressed my concerns.
